# Mesenchymal Stem Cell in Pancreatic Islet Transplantation

**DOI:** 10.3390/biomedicines11051426

**Published:** 2023-05-11

**Authors:** Serena Barachini, Letizia Biso, Shivakumar Kolachalam, Iacopo Petrini, Roberto Maggio, Marco Scarselli, Biancamaria Longoni

**Affiliations:** 1Department of Clinical and Experimental Medicine, University of Pisa, 56126 Pisa, Italy; 2Department of Translational Research and New Technologies in Medicine and Surgery, University of Pisa, 56126 Pisa, Italy; 3Aseptic Pharmacy, Charing Cross Hospital, Imperial College Healthcare NHS Trust, London W6 8RF, UK; 4Department of Biotechnological and Applied Clinical Sciences, University of L’Aquila, 67100 L’Aquila, Italy

**Keywords:** mesenchymal stem cells, transplantation, nanotubes, immunomodulation, pancreatic islets, exosomes

## Abstract

Pancreatic islet transplantation is a therapeutic option for achieving physiologic regulation of plasma glucose in Type 1 diabetic patients. At the same time, mesenchymal stem cells (MSCs) have demonstrated their potential in controlling graft rejection, the most fearsome complication in organ/tissue transplantation. MSCs can interact with innate and adaptive immune system cells either through direct cell-cell contact or through their secretome including exosomes. In this review, we discuss current findings regarding the graft microenvironment of pancreatic islet recipient patients and the crucial role of MSCs operation as cell managers able to control the immune system to prevent rejection and promote endogenous repair. We also discuss how challenging stressors, such as oxidative stress and impaired vasculogenesis, may jeopardize graft outcomes. In order to face these adverse conditions, we consider either hypoxia-exposure preconditioning of MSCs or human stem cells with angiogenic potential in organoids to overcome islets’ lack of vasculature. Along with the shepherding of carbon nanotubes-loaded MSCs to the transplantation site by a magnetic field, these studies look forward to exploiting MSCs stemness and their immunomodulatory properties in pancreatic islet transplantation.

## 1. Mesenchymal Stem Cells

Mesenchymal stem cells (MSCs) represent a cell source for maintaining homeostasis under physiological and pathological conditions in cell-based therapy [1]. Unlike other stem cells, the use of MSCs does not pose any ethical concern, and importantly, their therapeutic potentials known in pre-clinical animal disease models appear to have been fulfilled in clinical trials.

MSCs have been isolated from different biological tissues, such as adult bone marrow (BM) and adipose tissues, but also from pre-natal tissues, such as the umbilical cord and placenta, and are capable of extensive proliferation, multipotency and homing/migration [2]. MSCs were isolated for the first time by Friedenstein in mouse BM [3], and because they showed immunosuppressive functions, both in vitro and in vivo, they were initially described as hypoimmunogenic or ‘immune-privileged’; this property was thought to allow MSC transplantation through the main histocompatibility barriers.

However, recent studies have showed the production of antibodies against MSCs, with a subsequent rejection of allogenic donor MSCs mediated by the immune system, which led to reconsideration of MSCs as immune-privileged. Nonetheless, MSCs may still offer therapeutic functions through a brief ‘hit-and-run’ mechanism that protects them from recognition by the immune system and prolongs their survival in vivo, thereby improving clinical outcomes and preventing patient sensitization toward donor antigens. Furthermore, Davies and co-workers tested the differences of infusing autologous MSCs (from the host diabetic patient), or allogenic MSCs from healthy donors and demonstrated that there is no functional difference in terms of immunosuppressive activity, blood compatibility, or migratory capacity between the groups [4]. From a translational perspective, MSCs, either autologous or allogenic, appear to have evolved from immune-privileged to immune-evasive [5].

Type 1 diabetes mellitus (T1DM) is an autoimmune disease that leads insulin-producing β cells to death, caused by self-reactive T-cells, and represents one of the most frequent chronic diseases in children and adolescents, which incidence has increased in recent years. Since 1922, the administration of exogenous insulin has been the most crucial life-saving therapy for all patients with T1DM. However, chronic insulin treatment often fails to prevent long-term complications such as ketoacidosis, kidney failure, cardiovascular disease, neuropathy, and retinopathy. The only possibility of finding a definitive treatment for T1DM would be to provide a new source of β cells with the ability of sensing blood sugar levels and secreting insulin.

Pancreatic islet transplantation is a therapeutic option for treating T1DM, potentially avoiding the need for exogenous insulin. However, complications that can occur soon after the transplant are the acute loss of islets, which is caused by scarce vascularization, and hypoxic milieu, and these may lead to a poor transplantation outcome; moreover, it is estimated that at least 25% of the islet graft is lost during the transplantation procedure [6]. Moreover, despite immunosuppressive drugs, the immunological rejection may also contribute to graft failure. Some older studies showed that most of the patients resumed exogenous insulin treatment in a few years after transplantation [7]. However, recent studies report more optimistic results, estimating that insulin independence may be maintained by about 60% of the patients at five years post-transplant [7,8]. The ability of MSCs to communicate with innate and adaptive immune system cells either through direct cell-cell interaction or through their secretome, which contains numerous soluble growth and immunomodulatory factors (microRNA, transfer RNA, long noncoding RNA, growth factors, proteins, and lipids) [9], or via mitochondrial transfer may reduce the burden of immune rejection to achieve transplant tolerance [10]. In addition, it is relevant to stress that multiple injections of MSCs, administered systemically in the days following islet transplantation, have the ability to migrate to damaged sites and to contribute to tissue repairing [11,12]. It is to be considered that another option of treatment is described as simultaneous co-transplantation of pancreatic islets plus MSCs [13]. Hence, the immunomodulating, anti-inflammatory, and regenerating properties of MSCs might play a crucial role in aiding graft survival without requiring the use of immunosuppressive drugs, a concept whose importance has been previously described [14]. In this regard it has been shown that MSCs have an inhibitory effect on T-cell proliferation in vitro, and that they are able to slow down the rejection of allogeneic islet in a murine model of transplant, suggesting the intriguing possibility of using synergic cell therapy to prevent graft rejection [15]. These observations are especially attractive when considering the already established clinical use of MSCs [16,17,18]. Furthermore, in studies performed in a murine model of islet transplant, diabetic recipients showed reduced anti-donor IgG levels and a glucose tolerance similar to that of naïve nondiabetic mice [19].

Some authors have showed that the infusion of MSCs can decrease systemic inflammation levels [20,21], serum amylase [22], acinar cell apoptosis [23], and mortality [24] in rodent models of acute pancreatitis. Not all MSCs mechanism of action have been discovered; however, evidence suggests that the MSC secretome exerts a paracrine effect, playing a protective role in acute pancreatitis [25].

Organ engineering could be strategic for recreating a highly vascularized 3D microenvironment providing a multi-compartment structure with a dense vascular network improving the islets’ engraftment in T1DM. Recently, different groups have produced islet-like clusters/aggregates, and islet-like organoids, derived from induced pluripotent stem cells (iPSCs). This promising application, however, is still limited by scarce vasculogenesis around the engraftment, with a subsequent reduction in graft functionality [26]. For this reason, human stem cells with angiogenic potential in organoids, as a source of endothelial progenitors, could obviate the lack of vasculature. Recently, our group has isolated from human bone marrow (BM), a novel cell population of progenitors of the mesengenic lineage with angiogenic potential, named Mesangiogenic Progenitor Cells (MPCs) [27,28] that can generate exponentially growing MSC cultures in vitro. MPCs are tissue-specific and, in congruence with a previously reported population named *Pop#8* [29,30] and characterized by CD64*bright*CD31*bright*CD14*neg*CD45*dim*, were consistently detected exclusively in BM-mononuclear cells, giving rise to the isolation of MPCs under selective culture conditions [31,32]. Interestingly, the high expression in MPCs of CD31 and Nestin, expressed also in endothelial cells of small arteries and endosteal arterioles [33], suggest that these more primitive population could be considered as an endothelial lineages progenitor. Thus, MPCs could represent a valuable stem cell population for tissue engineering, where an important role in the establishment of successful engraftments is played by neo-vascularization.

## 2. Stressors Challenging Pancreatic Islet Graft

In T1DM patients, pancreatic islet transplantation may help to improve physiologic glycemic control by replacing the pancreatic β-cells lost due to an autoimmune attack [34]. However, hypoxia and oxidative stress cause a significant loss of grafted islets, and this represents a key challenge in islet transplantation.

Poor vascularization of grafted islets and the hypoxic milieu that can occur in the post-transplantation period may contribute to the acute loss of islets. Indeed, since the original report [35], technical improvements have refined the isolation procedure, control of cold ischemia and total ischemia time. Notably, 452 pancreas isolations assessed by a retrospective study have been correlated with clinical post-transplantation outcomes by Wassmer et al. [36]. Their data show that total and cold ischemia time, and organ removal time all adversely influence isolation of viable islets, and consequently the post-transplant outcome. Specifically, exceeding a total ischemia time of over 8 h reduces transplantation success. However, additional factors appear to affect islet transplantation outcomes. For example, a donor pancreas obtained from an ideal donor with a healthy lifestyle (age, body mass index, no alcohol or tobacco use, no history of high blood pressure, etc.) may allow the total ischemia time to exceed 12 h. Therefore, despite continuous improvement in the isolation technique, it seems that the success of islet transplantation remains linked to donor lifestyle and health conditions.

Oxidative stress is another factor affecting the morphology and function of isolated islets. Metabolic stress has been reported to increase the susceptibility of isolated pancreatic β cells to oxidative stress that is correlated with the production of reactive oxygen species (ROS) by reducing their antioxidative capacity [37]. Free radicals and ROS are actively produced in living systems in physiological conditions, but in relatively low levels. Therefore, when cellular antioxidant defense mechanisms in the transplanted islets are overwhelmed by an increased rate of intracellular ROS generation, this imbalance may lead to cell death. The cellular antioxidant defense mechanisms, including superoxide dismutase (SOD), catalase (CAT), and glutathione peroxidase (GPX), which maintain cellular homeostasis in normal conditions, are reduced in islet cells. Miki et al. [38] have shown that both CAT and GPX expression were significantly lower in pancreatic β- than α-cells, and the β/α-cell ratio decreased significantly after islet isolation and transplantation. Unsurprisingly, oxidative stress caused greater DNA damage in α-cells and significantly decreased the β-cell viability. These observations highlight the elevated susceptibility of pancreatic β-cell to oxidative stress as a leading cause of their loss.

Nutrients present in staple foods may affect islet cell viability and they might protect isolated pancreatic islets. Cultured pig islets were protected from warm and cold ischemia damage when pre-treated with glutamine, and this beneficial effect seems to be correlated with the intra-islet generation of glutathione (GSH) [39].

Interestingly, Nzuza et al. [40] have reported in vitro an antiapoptotic effect of naringin, which is derived from grapefruit, against protease inhibitors (PIs)-associated oxidative damage to pancreatic β-cells. PIs significantly reduce glucose-dependent insulin secretion in a concentration-dependent manner, and both naringin and glibenclamide were able to significantly reduce lipid peroxidation and SOD activities and increase GSH and adenosine triphosphate (ATP) levels in the PIs-treated cells. Furthermore, both compounds were able to significantly reduce caspase-3 and caspase-9 activities as well in the PIs-treated cells.

Sulforaphane (SFN), a natural isothiocyanate that derived from a glucosinolate contained in cruciferous vegetables, was shown to prevent pancreatic β-cell damage induced by cytokines downstream of the NF-kB signaling pathway, and restore insulin secretion in response to extracellular glucose. SFN increases sirtuin gene expression, a crucial modulator of mitochondrial function. This increased spare capacity and improved the electron flow, resulting in higher ATP turnover, thus an increase in the ATP/ADP ratio that guarantees continued exocytosis of insulin. Furthermore, SFN has been reported to decrease the levels of proinflammatory cytokines, including interleukin-1β (IL-1β), tumor necrosis factor-α (TNF-α), and interferon-γ (IFNγ) [41].

Recently, Keshtkar et al. [42] have shown in cultured human islets that nobiletin, a flavonoid from tangerine, increased antiapoptotic gene expression, such as *Bcl-2* and conversely decreased pro-apoptotic protein expression, such as BAX. In addition, a dose-dependent reduction in active caspase-3 was observed in the nobiletin group, indicating the inhibition of apoptosis and increasing the islet viability. The latter was assessed by fluorescein diacetate and propidium iodide staining at different concentrations of nobiletin, indicating that viable islets percentage was around 100% in the nobiletin-treated group, while in the control group the viability was reduced by 50%. Moreover, nobiletin enhanced the C-peptide and insulin indexes compared to the control group during 72 h of incubation, improving the islet function in the culture. Furthermore, there was an increment in insulin mRNA levels by nobiletin that correlated positively with the up-regulation of insulin secretion. The primary hypoxic marker, hypoxia inducible factor (HIF-1α), was expressed in the cultured islets in a similar manner in the control and nobiletin-treated groups at 24 h. In contrast, nobiletin reduced HIF-1α protein in the islets during 72 h of culture; however, other groups did not reflect this change in gene expression.

The relation between hypoxia and inflammation has been well described by Eltzschig and Carmeliet [43]. Hypoxia can cause a clinically relevant development of inflammation, as ischemia leads to an increment in the risk of inflammation in organ grafts, resulting in graft failure or rejection.

It is well known that not all successfully isolated viable islets will be transplanted. In a sample of five patients receiving six islet transplants, it was reported that at least 25% of the islet graft was lost during the surgery. This calculation was based on the level of C-peptide measured in circulation, indicating that in the post-transplantation period the islets get entrapped in clots, leading to delayed secretion of C-peptide [6]. Moreover, a subsequent early islet loss has been described as a consequence of excessive metabolic demand that could induce endoplasmic reticulum stress [44].

When MSCs are in a resting state, they tend to show immune homeostatic features shift towards an immunosuppressive phenotype [45]. This state can be further enhanced by the exposure to proinflammatory cytokines, such as TNF-α and IFNγ [46,47]. MSCs can therefore increase the levels of numerous anti-inflammatory agents [48], a role that is further potentiated by molecules such as Ras-related protein (Rap1) and IDO 1 [49,50]. Inflammatory signaling pathways, i.e., telomerase-associated protein Rap1/NF-kB pathway, regulate the paracrine function of MSCs [51]. This has been confirmed in a mouse model of allogenic transplantation in which the inhibition of NF-kB permitted a longer graft survival along with a decrease in inflammation [52]. Another study showed the potentials of MSCs as immunomodulators in the prevention of autoimmune attacks by autoreactive T-cells [53].

Because many factors are involved in challenging all procedures of islet transplantation and its successful engraftment, much can be still investigated to improve this therapeutic option.

## 3. MSC Immunomodulation and Advanced Medicinal Therapy

Several studies have investigated MSCs’ immunology and their potential to evade and/or influence the immune system of the host [54]. This is in relation to both immunosuppression and/or immunoprivilege. The former is due to MSCs’ ability to suppress recipient immune cells, while the latter is based on the low expression of the major histocompatibility complex (MHC) on the surface of MSCs. Moreover, in vitro MSCs express low levels of class I (HLA A, B, and C) MHC and no class II MHC, meaning that their use would not require HLA compatibility [55]. It is to be considered that within an inflammatory microenvironment, IFN-γ is crucial for driving MSCs to the anti-inflammatory state. In addition, exogeneous IFN-γ has been found to trigger MSC immunomodulatory properties [46]. BM-derived MSCs can be beneficial for immunomodulation in allotransplantation cell therapy. MSCs can inhibit the proliferation, activation, and function of T-cells directly by expressing immunomodulatory factors. The interaction between MSC and immune cells can further slow down proinflammatory T-cell responses. Consequently, in vivo strategy to pre-activate MSC with IFN-γ may be effective in reducing the inflammation in allotransplantation. In addition to the highly immunosuppressive phenotype, these MSCs have an improved homing ability, making them a very attractive tool for allotransplant cell therapy. Nevertheless, enhanced expression of MHC in allogeneic MSC plus IFN-γ may increase their immunogenicity and this could negatively impact allograft survival. Further studies aimed to reduce immunogenicity of the above MSCs phenotype may contribute to improve this therapeutic approach.

MSC immunomodulation mechanisms are not fully understood, but it is believed that both cell-to-cell contact and released trophic factors play a central role (Figure 1). In addition, MSCs can also be effective by migrating to the injured site and eliciting a therapeutic effect through immunomodulation and angiogenesis.

Numerous studies have shown that MSCs have a pleiotropic immunomodulatory effect. They impede ongoing immune processes and enhance tissue engraftment and β-cell viability. MSCs achieve this by creating a cytoprotective environment and releasing immunomodulatory molecules. Their effect influences all immune cells, including T lymphocytes and B lymphocytes, but also natural killers (NK), dendritic cells (DCs), and macrophages. The inhibition of T lymphocyte proliferation was shown to be mediated by the secretion of transforming growth factor-β (TGF β), hepatocyte growth factor (HGF), prostaglandin-E2 (PGE2), and the induction of tolerogenic activity of DCs. MSCs can modify cytokine release profiles of DCs, T-cells, and NK cells to promote an anti-inflammatory or tolerant phenotype. They can affect mature DC types, including DC1, by diminishing the secretion of TNF-α, modifying DC2 to promote IL-10 secretion, adjusting Th1 cells to decrease IFN-γ release, and provoking TH2 cells to increase IL-4 secretion. Additionally, they trigger an increase in the frequency of Tregs and a decrease in IFN-γ produced by NK cells [56] (Figure 1).

Soluble factors such as TGF-β, PGE2, HGF, indoleamine-pyrrole 2, 3-dioxygenase (IDO), nitric oxide (NO), IL-10 and histocompatibility antigen, class I, G (HLA-G5) contribute to the immunomodulatory effects by MSC. HLA-G belongs to HLA class Ib which could protect target cells from cytotoxic cell activities (i.e., NK cells). HLA-G has different phenotypes (G1 to G4 membrane attached and G5 to G8 soluble type) and soluble HLA-G secreted from MSC is one of the factors involved in protection of MSC from cytotoxic T lymphocytes-mediated killing [57]. PGE2 and NO are proposed as central factors stimulating T-cell suppression, while IDO can both facilitate the degradation of tryptophan, which are necessary for T-cell function, resulting in immunosuppression, and induce NK cell apoptosis. Other factors contribute also to MSC-induced immunomodulation, i.e., programmed cell death ligand 1 (PD-L1), FasL, and HGF [58]. Overall, MSCs have great potential for future clinical treatment due to their potent immunomodulatory and anti-inflammatory effects.

In recent years, the microenvironment has revealed novel targets, becoming an attractive field of research in drug development. The literature shows various examples of identification of the immune components and extracellular matrix of the microenvironment, such as fibronectin, and their respective correlation with tissue remodeling and repair, fibrosis and cell migration. For example, the presence of tumor heterogeneity, which is constantly evolving throughout the disease’s development and is also consequent to therapy, is a crucial point in the development of a microenvironment-specific treatment strategy. This heterogeneity creates the base for therapy evasion and resistance to the treatment, which are facilitated by multiple compensatory mechanisms and feedback loops [59]. The idea of exploiting the ability of MSCs to integrate into the microenvironment, as if they were a Trojan horse, and engineering them to increase their immunomodulatory potential has gained ground. The first studies in this direction have confirmed the feasibility of this strategy, and MSCs have been used as vehicles for the targeted delivery of immunostimulatory factors. There are different methods to guide MSCs to the transplantation site including target administration, magnetic guidance, genetic modification, cell surface engineering, in vitro priming and modification of the target tissue, and radiotherapeutic techniques [60]. Several papers have shown that when MSCs are administered systemically they distribute with preference to a wide plethora of organs, i.e., spleen and lungs, which results in therapeutic efficacy loss. One of the biggest challenges facing MSC therapies is enhancing their homing efficiency. Some of these strategies involve non-systemic homing, i.e., the targeted administration by magnetic guidance [61]. In particular, Vittorio et al. [62] investigated the ability of magnetic carbon nanotubes to guide MSCs intravenously in living rats into the target organ (liver) by using an external magnetic field. The authors demonstrated that the nanotubes interact with the MSCs without compromising their viability, proliferation rate, phenotype, or cytoskeletal conformation. These findings open up opportunities to utilize the magnetic properties of carbon nanotubes for the purpose of localizing stem cells in a target organ upon transplantation (Figure 2).

MSCs could be considered as advanced medicinal therapy when compared to drugs; however, their mechanism of action and tissue distribution in target diseases are not completely known. The current understanding is that their mechanism of action involves their ability to engraft, differentiate, and/or release paracrine signals, but the contribution of each property is unclear. This mechanism of action is described as a complicated network that triggers different reactions involving nearby cells to generate the adequate biological function for therapeutic effects. The exact role of the cells versus the released mediators in the therapeutic effect is still unknown and may be linked to the target disease and microenvironment.

Biomolecules released by MSCs have been observed to provide benefits beyond what is conveyed by transplanted cells alone. The unclear functions of MSCs and their related bioactive molecules pose a challenge in defining biomarkers to assess their mechanism of action and efficacy. Additionally, each disease’s microenvironmental peculiarity could differentially affect the biological functions of MSCs after transplantation. Kusuma et al. [63] suggest that the environment surrounding the MSCs is crucial to direct paracrine activity. Protein secretion can be modulated by a variety of extracellular cues, such as soluble factors, oxygen tension, and matrix-mediated physical and mechanical cues. The in vitro microenvironment contributes to creating the best MSC state for clinical applications by relying on secreted factors or exosome signaling improving the differentiation efficiency of MSCs within tissue-engineered constructs. It has been established that MSCs can release factors with the potential to guide the fate of other cells in that population [64] and leveraging these effects could offer further opportunities to drive MSC fate within biomaterial constructs.

Advances in biocompatible materials have strongly influenced MSC applications in tissue engineering by increasing cell survival and guiding cell differentiation in vivo. As a consequence, adult MSCs have been incorporated into various tissue engineering therapies, including cardiac repair, skeletal tissue repair, intervertebral disc repair, and cartilage and bone tissue regeneration.

While technologies continue to improve for creating high-quality MSC products for clinical use, both the advancements in bioengineering and the understanding of how the extracellular environment affect MSCs and how MSCs’ paracrine activity induce an immunomodulatory effect will be crucial for developing effective MSC therapies for clinical applications.

## 4. Exosomes and Its Potential in MSCs Vascularization

Exosomes are a subgroup of extracellular vesicles (EVs) produced by epithelial cells, tumor cells, B and T-cells, mast cells, and DCs and have been also found in blood, urine and breast milk [65]. At first, they were thought to serve as compartments for cellular debris removal, but in more recent years their importance in cell-to-cell communication has been recognized [66]. Exosomes have also gained considerable attention because they could be used as drug vehicles, as they are naturally non-immunogenic, therefore providing an unprecedented opportunity to enhance the delivery of encapsulated drugs to target cells [67]. In this regard, one must consider that the therapeutic effects of MSCs are attributed to their ability to secrete bioactive molecules influenced by either local microenvironment or MSC culture conditions. MSCs in culture secrete cytokines, micro-ribonucleic acid (miRNA), exosomes and microvesicles. The immunomodulatory mechanism of MSCs includes paracrine activity, cell-cell contact and interaction, mitochondrial transfer, MSC integration, and release of EVs [68]; this process is not equivalent to the mechanism involved in MSCs-induced tissue repair.

Hence, researchers are exploring MSC paracrine action in different clinical settings using an either conditioned medium (CM) or MSC-derived exosomes (EXOs), where these products regulate tissue responses depending on different microenvironments. The paracrine mechanism can enhance MSC therapeutic effects, where the composition of secretome can be modulated by priming the MSCs [69]. The therapeutic potential of MSCs is attributed to their ability to regulate immunity and other complex cellular and molecular mechanisms of action. The plasticity of MSCs in immunomodulation allows these cells to exert different immune effects according to different local conditions.

EXOs are enriched in numerous angiogenesis-related proteins, lipids and RNAs, including messenger RNA (mRNA) and miRNAs that could start diverse signaling pathways in endothelial cells inducing the expression of numerous trophic factors [70]. EXOs’ interaction with target cells involves ligands on the exosomal membrane surface binding to receptors on the target cell. This receptor-ligand interaction can induce one of three possible mechanisms: (1) exosomal membrane fusion with the cell’s plasma membrane, releasing cargo into the cytosol; (2) promoting of a downstream signaling cascade in the target cell; (3) exosome endocytosis, resulting in exosome fusion with the endosome’s membrane, and cargo release into the cytosol. Interestingly, it was shown that EXOs play an important role in neoangiogenesis. Several studies have shown that when EXOs are applied locally, they can induce neovascularization in several disease models [71,72,73]. Shabbir et al. [74], demonstrated that EXOs could enhance angiogenesis in vitro by inducing the expression of growth factors such as hepatocyte growth factor, insulin-like growth factor-1, nerve growth factor, stromal-derived growth factor-1 and vascular endothelial growth factor (VEGF), demonstrating the therapeutic potential of MSCs in wound healing. Sahoo et al. showed that human CD34+ stem cell-derived exosomes contain pro-angiogenic miRNAs, miRNA-126 and miRNA-130a, which may induce new blood vessels formation in ischemic tissue [75]. Moreover, human adipose-derived stem cells are enriched with miRNA-125a and miRNA-31 that could be transferred to vascular endothelial cells promoting angiogenesis. In addition, EXOs derived from human umbilical cord blood enhance the proliferation, migration, and tube-like formation of endothelial cells in vitro and in vivo inducing β-catenin activation in endothelial cells and increased the proliferating cell nuclear antigen (PCNA), cyclin-D3 and N-cadherin expression [76].

EXOs have the potential to enhance the efficiency and success rate of treatments involving transplanted pancreatic islets. This is because they can be involved in the angiogenesis and survival of the transplanted islets [77,78].

Among the complications of T1DM there is a significant increase in damage both to the central nervous system and to the kidney. Interestingly, EXOs have a strong therapeutic potential as a treatment of diabetic damages. Jiang et al. [79] have shown that EXOs from CM of human urine-derived stem cells injected into the kidneys of nephritic rats were able to prevent kidney damage from diabetes because they inhibit apoptosis and induce vascular regeneration. In addition, EXOs have been shown to repair damaged neurons after a stroke in animal models by repairing oxidative-induced damage in neurons and astrocytes so restoring cognitive function. This demonstrates the potential of exosome-based therapies in central nervous system damage [80]. EXOs can promote the proliferation of regulatory T-cells and the apoptosis of autoreactive T-cells, indicating that they could potentially be used to slow or stop the progression of T1DM, preserving pancreatic islet cells and insulin independence. EXO injections could be effective in preventing graft rejection without the need for immunosuppressive therapy. However, treatments would need to be started early, in relation to the disease progression, in order to retain insulin function.

Wen D et al. have shown that co-delivery of siFas and anti-miR-375 via EXOs from human BM can silence *Fas* and *miR-375* in human islets, improving their viability and function against inflammatory cytokines in humanized NOD *scid* gamma mice. Moreover, intravenous injection of BM-MSC and PBMC co-cultured exosomes can suppress immune activity by inhibiting PBMC proliferation and strengthen Treg function [81]. Thus, gene overexpression or modified EXOs may provide stronger immune regulation than untreated EXOs.

Currently, there are several challenges in the clinical application of EXOs, including the need for a standard method for exosome yield, further understanding of their components, and further exploration of immunoregulatory functions and long-term safety. However, despite these challenges, the advantages and potential of EXOs are attractive because of their low immunogenicity, long half-life [82], in vivo stability, and high delivery efficiency, making them safer and more efficient than stem cell therapy [83]. As a newer non-cellular biological therapy, EXOs have great potential.

## 5. Preconditioning of MSC as a Potential Therapeutic Strategy

Preconditioning of MSCs to promote quiescence is recently gaining popularity as it can deliver increased survival and regenerative potential upon transplantation. It is thought that functions of MSCs may be impaired during their isolation and cultivation for a long time in vitro [84]. Furthermore, once injected at their target location in the recipient, followed by their subsequent migration to the islet transplantation site, they confront an adverse microenvironment [85]. Therefore, strategies aimed to preserve the stemness of MSCs that allow them to cope with apoptotic signals consequent to inadequate structural integrity and function between graft and tissues should be set in order to improve their role in pancreatic islet graft success in terms of increasing their survival rate and reducing their death. The immunomodulating, anti-inflammatory, and regenerative secretions of MSCs have been linked to hypoxia, which enhances growth factors and anti-inflammatory molecules production [86]. Cellular adaptations to hypoxia depend on the transcription factor HIF, which remains inactive in high-oxygen condition and is activated under hypoxic conditions. During hypoxia, the HIF-1α activates a wide variety of genes implicated in cell adaptation pathways such as survival, angiogenesis, metabolism, and apoptosis [87]. Indeed, hypoxia represents one of the most common elements of tissue injury leading to activation of the HIF-1α, which induces transcription of angiogenic genes such as *VEGF* [88].

Stokes et al. [89] reported that HIF-1α potentiates β-cell survival after islet transplantation in both human and mouse islets. The decreased levels of HIF-1α impaired extracellular glucose-stimulated ATP generation and β-cell function, whereas increased levels of HIF-1α improved insulin secretion, glucose tolerance and β-cell function. The authors suggest that HIF-1α might be used as a therapeutic target for the β-cell dysfunction in type 2 diabetes mellitus (T2DM).

Interestingly, MSCs are usually incubated in 21% O_2_ (an important environmental factor) during expansion and in vitro experiments, but incubation of MSCs in 1% O_2_ leads to a decrease in apoptosis and in an increase in soluble factors secretion, such as VEGF-A and fibroblast growth factor (FGF-2) while retaining MSCs’ immunosuppressive abilities [90]. HIF-1α is detected in MSC lysate after 48 h of incubation in hypoxic conditions. Instead, MSCs incubated in normoxia and short-term exposure to hypoxia resulted in a significantly increased colony number compared to normoxia alone. In addition, MSCs incubated in hypoxia released increased levels of VEGF-A and FGF-2 compared to MSCs incubated in normoxia [91].

Furthermore, Touani et al. [92] have shown that pharmacological preconditioning with celastrol, a natural potent antioxidative plant used in traditional Oriental medicine, could increase the viability and functions of MSCs encapsulated within an injectable scaffold. The MSC paracrine function was preserved and improved as demonstrated by quantifying the proangiogenic growth factors released and the proliferation of human umbilical vein endothelial cells (HUVECs) co-cultured with encapsulated human MSCs. Furthermore, in a rat ischemia model, celastrol-treated cells, but not vehicle-treated cells, led to significantly increased vascularization in the peri-implant region after one week compared to the control. This occurred even after taking into consideration the fact that the scaffold might exacerbate hypoxia and nutrient depletion, which can have a negative effect on their viability and thus the therapeutic effects. A MSC therapy requires not only cell survival, but also efficient paracrine activity in target tissues. In fact, in ischemic tissues, the MSC paracrine activity is fundamental in promoting neoangiogenesis and subsequent tissue reperfusion. Effectively, in this system, the CM and the co-cultured cells showed that increases in cytokines VEGF-A, stromal cell-derived factor (SDF-1α), and FGF-2 are mainly involved in the process of re-vascularization, cell migration, and proliferation.

Pre-transplantation treatment by deferoxamine (DFO) increases mRNA expression and protein translation of protective factors prior to be subjected to insult at transplantation site through improved energy supply (ATP) before exposure to hypoxia [93]. This condition increases survival of donor endothelium, which not only supports revascularization but also facilitates delivery of nutrients to islets. A simple drug treatment with DFO along with induction of HIF-1α has shown increased gene expression changes that most likely contributed to decreased apoptosis at 24 h along with improved ATP [89]. These results were consistent with reports that DFO treatment improves outcomes for transplants in non-obese diabetic (NOD) mice and for allogeneic grafts [94]. Based on these observations, the authors conclude that HIF-1α acts as a protective factor in islet transplantation and is crucial for successful islet transplant outcomes [89].

It is important for MSCs to be able to expand outside their quiescent state, in order to be used in clinical settings. Therefore, a research group used a xenogen-free medium for in vitro amplification of MSCs [95]. MSC primed in this way could be able to secrete trophic factors, consequently increasing angiogenesis and reducing inflammation by fully integrating and differentiating into the regenerating site.

In another study conducted by Ferro et al. [96], it was shown that preconditioning of human BM-MSC by inducing complete starvation in a serum-free medium increases the secretion of trophic factors. This leads to an improvement in the early stage of fracture healing in diabetic mice, which is characterized by decreased proinflammatory cytokine expression, increased anti-inflammatory cytokine production, angiogenesis, and progenitor cell recruitment. As a result, there is an earlier formation of a soft callus, mineralization, and stabilization of the fracture. Preconditioning may be crucial for refining and defining new criteria for future MSC therapies.

## 6. Conclusions

MSCs show great promise for new medical treatments. Development of methodologies for handling of MSCs and their current and potential uses are of wide interest for research and medical practice.

Observing the effect of MSCs as regenerative as well as immunomodulatory tools helps the researchers better understand how disease and associated conditions develop. The immunomodulatory action of MSCs, regulated by cell-to-cell contact, by their secretome and by their interaction with immune cells, could control the progression of different inflammatory diseases as well as organ/tissue engraftment. Thus, the stem cell therapy has great potential to benefit not only pancreatic islet transplantation efficacy but also many other patients who suffer from spinal cord injuries, Parkinson’s or other neurodegenerative diseases, stroke and bone disease.

This review highlights several strategies such as the right conditioning of MSCs to influence their viability and their therapeutic efficacy, the possibility of taking advantage of MSCs-derived exosomes and that of shepherding carbon nanotubes-loaded MSCs to the transplantation site by an external magnetic field. Future work should point to new routes of preconditioning to enhance MSC efficacy and to ensure their intact secretome activity and therapeutic efficacy, especially with regard to clinical translation.

## Figures and Tables

**Figure 1 biomedicines-11-01426-f001:**
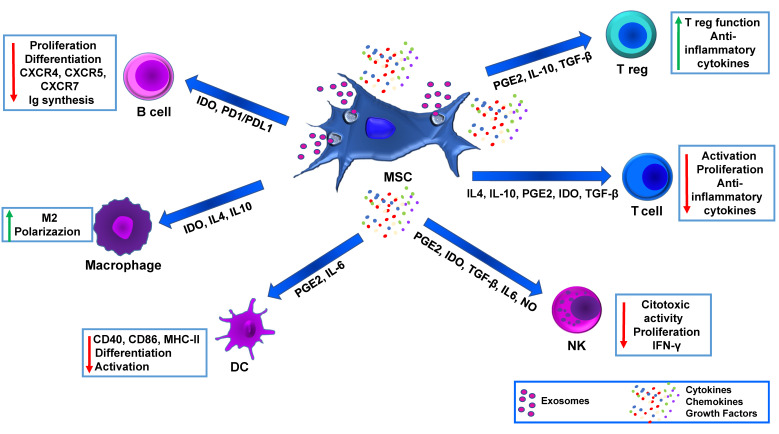
MSC immunomodulatory effects. MSC can exert an immunomodulatory effect on target cells through several mechanisms including biologically active factors (cytokine, chemokines, growth factors) and exosomes which contain lipids, proteins, miRNA. Red arrows indicate down-regulation, green arrows indicate up-regulation.

**Figure 2 biomedicines-11-01426-f002:**
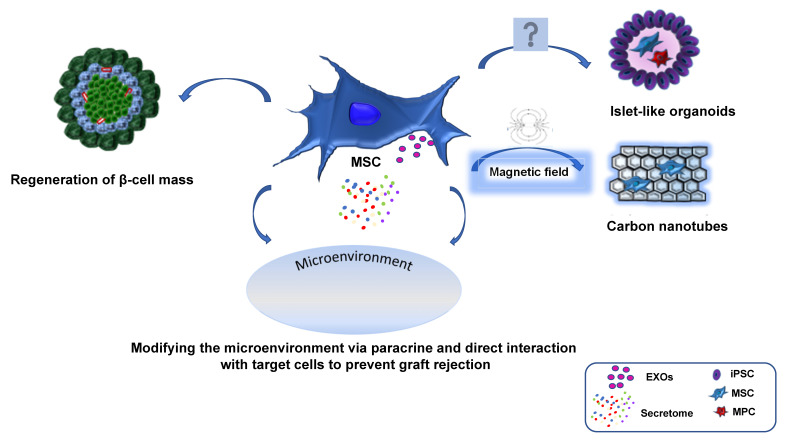
Role of MSC in islet transplantation. MSCs, either autologous or allogenic, have the potential to enhance the efficiency and success rate of treatments involving transplanted pancreatic islets through several mechanisms including regeneration of β-cell mass, generation of vascularized islet-like organoids and modifying the microenvironment via paracrine effects of their secretome and EXOs. In addition, MSC-loaded magnetic carbon nanotubes can play an important role in biomedical application enhancing the homing of stem cells towards the target organ. MSC: mesenchymal stem cells, MPC: mesangiogenic progenitor cells, iPSC: induced pluripotent stem cells, EXO: exosome-derived MSCs.

## Data Availability

Not applicable.

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
