# Peer review of "Mesenchymal Stem Cell in Pancreatic Islet Transplantation"

_biomedicines, 2023, doi:10.3390/biomedicines11051426_

Round 1
Reviewer 1 Report
This review article shows the roles of MSCs and usefulness of MSCs for islet transplantation, but does not mention about which MSCs (auto, allo, or xeno) and How (co-transplantation, co-culture, etc.) to use. So I cannot imagine how the authors deal with MSCs in islet transplantation. Please shows the therapeutic images in resubmission.
No comments because this reviewer is a native English speaker. At least, I can understand what the author described.
Author Response
Dear Reviewer,
Thank you for taking the time to review my manuscript and for providing invaluable feedback. I am grateful for your positive comments on the overall organization and content of the review.
I would like to express my sincere gratitude for your comments on my manuscript. Your feedback has been helpful in enhancing the significance of the manuscript and expanding on the missing topic, both in text and in the legend of figure2, regarding the autologous or allogenic MSC.
- Lindsay C. Davies, Jessica J. Alm, Nina Heldring, Guido Moll, Caroline Gavin, Ioannis Batsis, Hong Qian, Mikael Sigvardsson, Bo Nilsson, Lauri E. Kyllonen . Type 1 Diabetes Mellitus Donor Mesenchymal Stromal Cells Exhibit Comparable Potency to Healthy Controls In Vitro . Stem Cells Translational Medicine, Volume 5, Issue 11, November 2016, Pages 1485–1495, https://doi.org/10.5966/sctm.2015-0272
Once again, I thank you for your valuable comments and suggestions, which have helped to enhance the quality and relevance of the manuscript. I hope that you find the revised version to be satisfactory and look forward to any further feedback you may have.
Reviewer 2 Report
In this review article, the authors discussed the reasons of low efficacy of pancreatic islet transplantation and beneficial effect of Mesenchymal Stem cells (MSC) treatment. The showed that MSC could Influence the microenvironment of transplanted pancreatic islets. By using carbon nanotubes, MSC could be guided to the sites of cell transplantation and create preferable microenvironment for transplanted islets by reducing inflammation and improving vascularization. Authors also discuss extra cellular vesicles (EV) for protection for hypoxia-caused apoptosis.
This review indicates potential benefits of using nanotube guiding injection of MSC and also EV for clinical treatment. It is well written, however, there are few corrections of text and references required.
1. Page 2
Authors wrote that ‘Some older studies showed that most of the patients resumed exogenous insulin treatment in a few years after transplantation.’ References should be added.
2. Page 2
Authors wrote that ‘Hence, the immunomodulating, anti-inflammatory and regenerating properties of MSC might play a crucial role in aiding graft survival, without requiring the use of immunosuppressive drugs, a concept whose importance has been previously described.’
It could be informative to add comments and references to some previous preclinical studies showing beneficial effect of co-infusion of syngeneic MSC on allogeneic pancreatic islet transplantation. (such as ref Jacobson S. Ann N Y Acad Sci 2008 Dec;1150:213-6. doi: 10.1196/annals.1447.042. and Takahashi T. Stem Cells Transl Med. 2014 Dec;3(12):1484-94. doi: 10.5966/sctm.2014-0012.)
3. Page 5
The first paragraph of 3. MSC-immunomodulation and advanced medicinal therapy, there was no reference cited. Please add references.
Furthermore, MSC express HLA class I (HLA A, B and C). This expression became stronger by stimulation with IFN-gamma. MSC are not homogenous, therefore, there could be contaminated some cells which could express HLA class II especially after IFN-gamma stimulation. Authors should consider these facts when describing the cell surface molecules.
4. Page 5
Authors wrote that ‘histocompatibility antigen, class I, G (HLA-G5)’. This should be mentioned in a different way.
HLAG belongs to HLA class Ib which could protect target cells from cytotoxic cell activities (for example NK cells). HLA G has different phenotypes (G1-G4 membrane attached and G5-G8 soluble type). Please see reference below where HLAG expression on MSC is described.
Stem Cells. 2008 May; 26(5): 1275–1287. doi:10.1634/stemcells.2007-0878.
5. Page 8
Authors wrote that ‘Exosomes are extracellular vesicles (EVs) …’. This can be corrected by addiing underlined part: ‘Exosomes are a subgroup of extracellular vesicles (EVs)…’
The definition of EV is stated in the reference below. Please consider.
J Extracell Vesicles. 2018; 7(1): 1535750. Published online 2018 Nov 23. doi: 10.1080/20013078.2018.1535750
6. Page 9
‘However, despite these challenges, the advantages and potential of EXOs are attractive because of their low immunogenicity, long half-life, in vivo stability, and high delivery efficiency, foreseeing them to be safer and more efficient than stem cell therapy.’
The underlined part is the copy from the summary part of another review article (ref 72). There is no clear description of ‘long half-life’ in the cited article, either. Please cite another article which clearly state how long the half-life of EVs is, or describe the expected half-life of EV in vivo with an original reference.
7. Page 11
Authors wrote in conclusions as ‘Many people might benefit from stem cell therapies including those with spinal cord injuries, T1DM, Parkinson’s or other neurodegenerative disease, heart disease, stroke and bone disease.’
This sentence should be continued to the previous paragraph instead of making a new paragraph. Also it would be easy to understand to rephrase it as below (for example).
“Thus, the stem cell therapy has great potential to benefit not only pancreatic islet transplantation efficacy but also many other patients who suffer from spinal cord injury….”
Good. one typo was noted.
Author Response
Dear Reviewer,
Thank you for taking the time to review my manuscript and for providing invaluable feedback. I am grateful for your positive comments on the overall organization and content of the review.
I appreciate your suggestion regarding the inclusion of more relevant sources on bibliography of the paper and I have revised the manuscript according to your indications to ensure that the sources I added are appropriately highlighted and discussed.
Once again, I thank you for your valuable comments and suggestions, which have helped to enhance the quality and relevance of the manuscript. I hope that you find the revised version to be satisfactory and look forward to any further feedback you may have.
- Page 2
The Reference showing that most of the patients resumed exogenous insulin treatment in a few years after transplantation has been added to the text:
- Ryan EA, Paty BW, Senior PA, et al. Five-year follow-up after clinical islet transplantation. Diabetes 2005;54(7):2060-9, doi:10.2337/diabetes.54.7.2060
- Page 2
Comments and references including some previous preclinical studies showing beneficial effect of co-infusion of syngeneic MSC on allogeneic pancreatic islet transplantation have been added to the text:
- Jacobson S, Kumagai-Braesch M, Tibell A, et al. Co-transplantation of stromal cells interferes with the rejection of allogeneic islet grafts. Ann N Y Acad Sci 2008;1150(213-6, doi:10.1196/annals.1447.042
- Le Blanc K, Rasmusson I, Sundberg B, et al. Treatment of severe acute graft-versus-host disease with third party haploidentical mesenchymal stem cells. Lancet 2004;363(9419):1439-41, doi:10.1016/S0140-6736(04)16104-7
- Bernardo ME, Locatelli F, Fibbe WE. Mesenchymal stromal cells. Ann N Y Acad Sci 2009;1176(101-17, doi:10.1111/j.1749-6632.2009.04607.x
- Bernardo ME, Fibbe WE. Safety and efficacy of mesenchymal stromal cell therapy in autoimmune disorders. Ann N Y Acad Sci 2012;1266(107-17, doi:10.1111/j.1749-6632.2012.06667.x
- Takahashi T, Tibell A, Ljung K, et al. Multipotent mesenchymal stromal cells synergize with costimulation blockade in the inhibition of immune responses and the induction of Foxp3+ regulatory T cells. Stem Cells Transl Med 2014;3(12):1484-94, doi:10.5966/sctm.2014-0012
- Page 5
References on MSC-immunomodulation have been added to the text including the role of IFN-γ and its contribute to drive MSCs to the anti-inflammatory state:
- Yang G, Fan X, Liu Y, et al. Immunomodulatory Mechanisms and Therapeutic Potential of Mesenchymal Stem Cells. Stem Cell Rev Rep 2023;1-18, doi:10.1007/s12015-023-10539-9
- Krampera M, Cosmi L, Angeli R, et al. Role for interferon-gamma in the immunomodulatory activity of human bone marrow mesenchymal stem cells. Stem Cells 2006;24(2):386-98, doi:10.1634/stemcells.2005-0008
- Carvalho AS, Sousa MRR, Alencar-Silva T, et al. Mesenchymal stem cells immunomodulation: The road to IFN-γ licensing and the path ahead. Cytokine Growth Factor Rev 2019;47(32-42, doi:10.1016/j.cytogfr.2019.05.006
- Page 5
The reference on HLAG has been added in the text. In addition we have added the role of soluble HLAG secreted from MSC, a factor involved in protection of MSC from cytotoxic T lymphocytes-mediated killing.
- Stem Cells. 2008 May; 26(5): 1275–1287. doi:10.1634/stemcells.2007-0878
- Page 8
The sentence in the page 8 has been corrected adding “Exosomes are a subgroup of extracellular vesicles (EVs)” and the reference has been added :
- Théry C, Witwer KW, Aikawa E, et al. Minimal information for studies of extracellular vesicles 2018 (MISEV2018): a position statement of the International Society for Extracellular Vesicles and update of the MISEV2014 guidelines. J Extracell Vesicles 2018;7(1):1535750, doi:10.1080/20013078.2018.1535750
- Page 9
The reference has been added to the text demonstrating that exosomes derived from bone-marrow MSCs have a long half-life in a murine breast cancer model:
- Gomari H, Forouzandeh Moghadam M, Soleimani M, et al. Targeted delivery of doxorubicin to HER2 positive tumor models. Int J Nanomedicine 2019;14(5679-5690, doi:10.2147/IJN.S210731
- Page 11
The sentence in Conclusion: “Many people might benefit from stem cell therapies including those with spinal cord injuries, T1DM, Parkinson’s or other neurodegenerative disease, heart disease, stroke and bone disease” has been rephrased and moved to the previous paragraph.
Round 2
Reviewer 1 Report
Probably, the authors aim to show the usefulness of CO-TRANSPLANTATION of MSCs in islet transplantation. If so, the title should be changed to "Co-transplantation of mesenchymal stem cells in islet transplantation". Also change the text following the title change (includes "co-transplantation of MSCs..." in some parts). If not, please show how (co-transplantation or co-culture) to use MSCs in islet transplantation in the text.
Author Response
Dear Reviewer,
I apologize for my oversight in failing to include an important clarification in my previous communication regarding co-transplantation of MSCs. It was not my intention to leave out this crucial information, and I apologize for any confusion or inconvenience this may have caused.
In chapter 1 we added the following sentence in red: “In addition, it is relevant to stress that multiple injections of MSCs, administered sistemically in the following days from islet transplantation, have the ability to migrate in damaged sites and to contribute in tissue repairing. It is to be considered that another option of treatment is described as simultaneous co-transplantation of pancreatic islets plus MSCs”. We hope that this may clarify the point you suggested about MSCs administration.
To support the above point, we added the following references:
- Biancamaria Longoni 1, Erzsebet Szilagyi, Paola Quaranta, Giacomo Timoteo Paoli, Sergio Tripodi, Serena Urbani, Benedetta Mazzanti, Benedetta Rossi, Rosa Fanci, Gian Carlo Demontis, Pasquina Marzola, Riccardo Saccardi, Marcella Cintorino, Franco Mosca. Mesenchymal stem cells prevent acute rejection and prolong graft function in pancreatic islet transplantation. Diabetes Technol Ther. . 2010 Jun;12(6):435-46. doi: 10.1089/dia.2009.0154 PMID: 20470228
- Koehler N, Buhler L, Egger B, Gonelle-Gispert C. Multipotent Mesenchymal Stromal Cells Interact and Support Islet of Langerhans Viability and Function. Front Endocrinol (Lausanne). 2022 Feb 9;13:822191. doi: 10.3389/fendo.2022.822191. PMID: 35222280; PMCID: PMC8864309
Thank you for your understanding, and do not hesitate to contact us if you require any further information or clarification.
Kind regards,
Biancamaria Longoni
Round 3
Reviewer 1 Report
The authors revised the reviea manuscript following the reviewer's comments.